# *Microvirga sesbaniae* sp. nov. and *Microvirga yunnanensis* sp. nov., Pink-Pigmented Bacteria Isolated from Root Nodules of *Sesbania cannabina* (Retz.) Poir.

**DOI:** 10.3390/microorganisms12081558

**Published:** 2024-07-30

**Authors:** Nan Shi, Teng He, Huifang Qin, Ziye Wang, Shenghao You, Entao Wang, Guoli Hu, Fang Wang, Miao Yu, Xiaoyun Liu, Zhenyu Liu

**Affiliations:** 1Key Laboratory of Microbial Diversity Research and Application of Hebei Province, Engineering Research Center of Microbial Breeding and Conservation, Hebei Province, Institute of Life Science and Green Development, Hebei University, Baoding 071002, China; eshishi@126.com (N.S.); hett19960610@163.com (T.H.); qinhf1996@163.com (H.Q.); wangziye@psych.ac.cn (Z.W.); youshenghao@outlook.com (S.Y.); hgl0860@126.com (G.H.); yumiaosms@163.com (M.Y.); 2Affiliation Departamento de Microbiología, Escuela Nacional de Ciencias Biológicas, Instituto Politecnico Nacional, Mexico City 11340, Mexico; entaowang@yahoo.com.mx; 3Key Laboratory of State Forestry Administration for Biodiversity Conservation in Southwest China, Southwest Forestry University, Kunming 650224, China; fangerfriend2021@163.com; 4Institute of Agro-Resources and Environment, Hebei Academy of Agriculture and Forestry Sciences, Shijiazhuang 050051, China

**Keywords:** *Microvirga*, *Sesbania cannabina*, carotenoids, symbiotic nodules

## Abstract

Four pigment-producing rhizobial strains nodulating *Sesbania cannabina* (Retz.) Poir. formed a unique group in genus *Microvirga* in the phylogeny of a 16S rRNA gene and five housekeeping genes (*gyrB*, *recA*, *dnaK*, *glnA*, and *atpD*) in a genome analysis, phenotypic characteristics analysis, and chemotaxonomic analysis. These four strains shared as high as 99.3% similarity with *Microvirga tunisiensis* LmiM8^T^ in the 16S rRNA gene sequence and, in an MLSA, were subdivided into two clusters, ANI (genome average nucleotide) and dDDH (digital DNA–DNA hybridization) which shared sequence similarities lower than the species thresholds with each other and with the reference strains for related *Microvirga* species. The polar lipids elucidated that phosphatidylcholine (PC), phosphatidylethanolamine (PE), phosphatidylglycerol (PG), and cardiolipin were the main components for strain SWF67558^T^ and for strain HBU65207^T^, with the exception of PC. SWF67558^T^ and HBU65207^T^ strains had similar predominant cellular fatty acids, including C16:0, C18:0, summed feature 2, and summed feature8, but with different contents. In addition, all the four novel strains produced pink-pigment, and the main coloring material extract from strain SWF67558^T^ was identified as zeaxanthin, which presented antioxidant ability and reduction power. With all the phylogenetic and phenotypic divergency, we proposed these pink-pigmented symbiotic bacteria as two novel species, named *Microvirga sesbaniae* sp. nov. and *Microvirga yunnanensis* sp. nov., with SWF67558^T^ (=KCTC82331^T^=GDMCC1.2024^T^) and HBU65207^T^ (=KCTC92125^T^=GDMCC1.2023^T^) as the type strains, respectively.

## 1. Introduction

Rhizobia are soil bacteria composed of none-spore-forming, Gram-staining negative, aerobic rods that could form root- and/or stem-nodules with legume plants and reduce N_2_ into NH_3_ as a nitrogen nutrient to plants [1,2,3,4,5]. Extensive studies on the diversity and phylogeny of rhizobia have been explored in the past decades, about 200 species within 20 genera have been described for these symbiotic bacteria, and most of them are classified in the genera *Rhizobium*, *Mesorhizobium*, *Sinorhizobium* (*Ensifer*) and *Bradyrhizobium* of *Alphaproteobacteria* (alfa-rhizobia), or *Burkholderia* and *Cupriaviduas* of *Betaproteobacteria* (beta-rhizobia) [1,5,6]. These diverse rhizobia are widely distributed in different geographic regions in association with their host plants [4,5,6]. The host specificity determined by the legume’s preference for the symbiosis (nodulation) genes in rhizobia and the chromosome background’s determined adaptation ability of rhizobia have shaped the biogeographic patterns of rhizobia [3,6,7]. So, a single legume species could form a nodule symbiosis with distinct rhizobial populations or species in different geographic regions or soil types, like the cases of common bean and soybean [7,8].

Covering about 60 accepted species, *Sesbania* is the only genus in the legume tribe Sesbanieae, which are native to tropical and subtropical regions and are distributed in diverse habitats around the world, from aquatic to semi-arid ecosystems [9]. Many species in this genus are good green manure plants because of their formation of root and/or stem nodules with rhizobia for nitrogen fixation [10,11,12]. Diverse rhizobia nodulating with *Sesbania* species have been described, including *Azorhizobium caulinodans*, *Azorhizobium doebereinerae*, *Rhizobium huautlense*, *Sinorhizobium sesbaniae*, and *Sinorhizobium alkalisoli*, as well as strains in other species of *Agrobacterium*, *Bradyrhizobium*, *Mesorhizobium*, *Neorhizobium*, and *Rhizobium* [11,12,13,14,15,16,17,18,19]. In China, several *Sesbania* species have been recorded as native or introduced plants, such as *S. javanica*, *S. sesban*, *S. bispinosa*, *S. grandiflora*, and *S. cannabina*, of which only *S. cannabina*-associated rhizobia in the southern area have been studied [10,16,18,20].

As an annual semi-shrub effectively nodulating with rhizobia, *S. cannabina* (Retz.) Poir. is a widely planted green manure in tropical regions of China (eForas.org), based on its high efficient symbiotic nitrogen fixation [21] and strong resistance to abiotic stresses, such as salinity, waterlogging, and drought [22]. Currently, it has been widely introduced to the temperate region in northern China. Although diverse rhizobial species were evidenced as the microsymbionts for *S. cannabina*, highly conserved symbiotic genes have been detected in the *S. cannabina*-nodulating bacteria, and their geographic distribution was shaped by the host and soil properties [10]. These findings implied the possibility that lateral transfer of symbiosis genes played an important role for the diversification of *Sesbania* rhizobia [23,24,25]. Therefore, it could be hypothesized that more novel rhizobia exist in different regions associated with *S. cannabina.*

For searching effective rhizobia as an inoculant to improve the production of *S. cannabina*, root nodules were collected from a plant grown in Yunnan, and several rhizobia isolated from these nodules were identified as members of the genus *Microvirga* based on the phylogeny of 16S rRNA genes (our unpublished data). The genus *Microvirga* was established by Kanso and Patel [26] and amended by Weon et al. [27], which covered a total of 25 valid species up to date (http://www.bacterio.net/microvirga.html, 29 July 2024). In this genus, *M. ossetica*, *M. lupini*, *M. lotononidis*, *M. vignae*, *M. zambiensis*, and two invalid published species, *M. tunisiensis* and *M. calopogonii*, were nodule-inducing, nitrogen-fixing microsymbionts [23,24,28,29,30,31] associated with *Listia angolensis*, *Lupinus texensis*, *Lupinus micranthus*, *Lupinus luteus*, *Vicia alpestris*, *Calopogonium mucunoides*, and *Vigna unguiculata*, [23,28,29,30,31]. Considering that the isolation of *Microvirga* from *S*. *cannabina* evidenced a new symbiotic association between legume plants and rhizobia, we performed a systematic characterization on the *Microvirga* isolates from *S. cannabina* to clarify their species affiliation. Based on the results, two novel species, *Microvirga sesbaniae* and *Microvirga yunnanensis* were described, with SWF67558^T^ and HBU65207^T^ as the type strains, respectively.

## 2. Materials and Methods

### 2.1. Rhizobial Isolation and Genus Screening of the Isolates

For investigating the rhizobial resources of the legume plants in genus *Sesbania*, root nodules were collected from *Sesbania cannabina* (Retz.) Poir. grown in Santaixiang, a town in Manshi City (N24°19′, E98°24′, 870 m), and Yuanjiang, a town in Yuxi City (N23°35′, E101°53′, 342 m), which were separated by a straight distance of 350 km in the subtropical region of Yunnan Province. A total of 231 bacterial isolates were obtained from the nodules with the standard methods by streaking on the YMA plates [32]. All the isolates were cultured on the yeast–mannitol agar (YMA) [32] at 28 °C for 3–5 d and stored at −80 °C in a YM broth supplied with 20% (*w*/*v*) glycerol.

For screening the genus affiliation of the isolates, genomic DNA was extracted from each isolate by a guanidine isothiocyanate method using a FastPure Bacteria DNA isolation Mini kit (Nanjing Vazyme Biotech Co., Ltd. Nanjing, China) as suggested by the manufacturer. The DNA extract was used as a template for amplifying the 16S rRNA gene by PCR with the described protocol and primers fD1 (5′-AGA GTT TGA TCC TGG CTC AGA-3′) and rD1 (5′-AAG GAG GTG ATC CAG CC-3′) [33]. All the amplicons were sequenced with the commercial service of Beijing Genomics Institute (BGI, Beijing, China) using the same primers for PCR. The acquired sequences were blasted in the NCBI database to search the most closely related sequences, which were multiple aligned with the package Clustal X [34] to determine the genus affiliation of the isolates by their similarity with those from defined bacterial species, using the threshold of 95% similarity. As a result, four strains SWF67558^T^, HBU65207^T^, HBU67655, and HBU67692 from both sampling sites were identified as *Microvirga* (results not shown), and they were used in the subsequent analyses.

### 2.2. BOX-PCR for the Microvirga Strains

BOX-PCRs were performed to discriminate the four novel *Microvirga* strains and the references strain *Microvirga tunisiensis* LmiM8, using the extracted genomic DNA as template, with the primer BOXAIR (5′-CTA CGG CAA GGC GAC GCT GACG-3′) [35] and the protocol described by Liu et al. [36]. An aliquot of 5 µL PCR product for each sample was separated by electrophoresis in a 1.5% (*w*/*v*) agarose gel (15 cm long) at 150 V for 4 h. The electrophoresis patterns were visualized under UV light after staining with ethidium bromide.

### 2.3. Genome Sequence Analyses

The genomes of the four tested strains were sequenced with the commercial service of BGI (Shenzhen, China). A pair-ended library with insert sizes ranging from 350 bp and an SMRT bell library with insert sizes of 10 kb were constructed. The qualified libraries were sequenced on PacBio RSII and DNBSEQ platform. The program Pbdagcon (v2.0) (https://github.com/PacificBiosciences/pbdagcon) was used for self-correction. The corrected reads were assembled using the Canu (v1.5), Falcon (v3.0), and SMRT Analysis (v2.3.0). Moreover, Genome Analysis ToolKit 4 (gatk.broadinstitute.org/) and SOAP software packages (v1.2) were used to check the base corrections for improving the accuracy of the genome sequences and to check for the presence of any plasmids. As references of related species, complete or draft genome sequences of strains *M. tunisiensis* LmiM8^T^, *Microvirga calopogonii* CCBAU65841^T^, and *Microvirga lotononidis* WSM3357^T^ were extracted from the database. The annotation of the genome was conducted according to the RAST (v2.0) protocol and Swiss-Prot [37]. Additionally, the tRNA genes in the genome were predicted using tRNAscan-SE [38], while the rRNA genes were detected through RNAm-mer (v1.2) [39]. The gene annotation was conducted using the online websites of RAST (v2.0) (https://rast.nmpdr.org/) and EggNOG-mapper2 (v2.1) (http://eggnog-mapper.embl.de/).

The microbial polyketide and non-ribosomal peptide gene clusters [40] for the predicted number of polyketide synthase (PKS)- and non-ribosomal peptide synthetase (NRPS)-encoding genes were annotated by using the antiSMASH database (v7.0) (https://antismash.secondarymetabolites.org/), and only the best hit for each protein was kept. The genomes of *M. tunisiensis* LmiM8, *M. calopogonii* CCBAU65841, and *M. lotononidis* WSM3557 were used for genomic comparison.

The average nucleotide identities (ANIs, v2.1.2) among the genome sequences of the four tested strains and the reference strains were calculated online (www.ezbiocloud.net/tools), and the average amino acid identity (AAI, v1.2.1) values were also calculated online (http://leb.snu.ac.kr/ezaai). The average genomic identity of orthologous gene sequences (AGIOS) between the compared genomes was determined through OrthoANI software (v2.1.4) using the Needleman–Wunsch algorithm global alignment technique [41].

The reference strains were selected according to their similarity to the 16S rRNA gene and housekeeping genes with the four tested strains, and their genome sequences were extracted from GenBank database (http://www.ncbi.nlm.nih.gov/genbank/). In addition, digital DNA–DNA hybridization (DDH) was performed by the Genome-to-Genome Distance Calculator (v2.1) (http://ggdc.dsmz.de/distcalc2.php), based on the recommended BLAST+ alignment tool and formula 2 (identities/HSP length) [42]. The phylogeny of 120 core gene sequences was constructed via UBCG software (v3.0) [43], and the genome graphical circular map was obtained using the CGview tool (V2.0.3) [44].

### 2.4. Phylogenies of House-Keeping Genes

Currently, MLSA is used as an essential method to define rhizobial species [13,28], and the housekeeping genes, *atpD*, *recA*, *dnaK*, *gyrB*, and *glnA*, have been recommended for MLSA [45,46], which could compensate for the incapability of the 16S rRNA gene to distinguish the closely related rhizobial species. To compare the results of the genomic sequence analysis with the phylogenies of the house-keeping genes, a simple and popular method for genus and species of bacteria was used [15,47]; the sequences of the 16S rRNA genes and the five housekeeping genes were extracted from the genome of each of the four strains SWF67558^T^, HBU65207^T^, HBU67655, and HBU67692. The acquired 16S rRNA gene and house-keeping gene sequences were blasted in the GenBank database (http://www.ncbi.nlm.nih.gov/genbank/) for extracting the sequences of related reference strains, which were all compared with the related sequences in the EzTaxon server [48]. The acquired sequences in this study were aligned with the related reference sequences extracted from GenBank by using Clustal X [27]. Maximum likelihood (ML) and neighbor joining (NJ) phylogenetic trees were constructed using Mega 7.0 software [49] and were bootstrapped with 1000 pseudo-replicates. The optimum substitution model GT–GAMMA was used for the ML method.

### 2.5. Phenotypic Characteristics

The tested strains were cultured on YMA plates at 28 °C for 2–3 days, and then Gram staining was performed, and cell morphology was observed [50]. To observe cells motility, a light microscope with the hanging-drop method was used, and flagellum morphology was determined via transmission electron microscopy (Tecnai G2 F20 S-TWIN, FEI, Hillsboro, AL, USA), using cells grown on TY agar at 28 °C for 2 days. The utilization of carbon sources was assessed using Biolog GN2 microplates (Biolog, Hayward, CA, USA) and API 20NE (bioMérieux, Lyons, France) according to the manufacturers’ instructions by three repetitions, and the oxidase activity was determined with a moistened paper containing p-phenylenediamine by three repetitions, all of the tests were laid out by the YMA medium as the controls [51]. As described [51], salt tolerance was tested with YMA plates supplied with 1.0 to 5.0% (*w*/*v*) of NaCl, with an interval of 1%, and incubated at 28 °C. The pH range for growth was tested on YMA plates with pH 4.0 through 11.0, with an interval of 1 unit, and the temperature ranges for growth (at 4, 10, 28, 37, and 45 °C) were determined in YMA plates at pH 7.0 by incubating for 5 days. The antibiotic resistance of the tested strains were estimated on YMA plates supplied with 300, 100, 50, and 5 μg mL^−1^ of ampicillin, tetracycline, kanamycin, streptomycin, and chloramphenicol, respectively [51].

### 2.6. Chemotaxonomic Analyses

The polar lipids of strain SWF67558^T^ and HBU65207^T^ were extracted according to the method described previously [52], and the characteristic conditions of the polar lipids of the strain SWF 67558^T^ and HBU65207^T^ accord with the basic standards of the *Microvirga* species [32]. Briefly, 200 mg of wet cell pellets were used for the extraction of polar lipids, and the extracts were examined via a two-dimensional, thin-layer chromatography (TLC), using the mixtures of chloroform/methanol/water and chloroform/methanol/glacial acetic acid as the developing solvent. Then, the polar lipid spots were stained using iodine, molybdenum blue, and ninhydrin as the spray-developing reagents [53]. For fatty acid analysis, cell harvesting and fatty acids extraction were performed as previously reported [54]. The extracts were analyzed by gas chromatography (Aglient Technologies Co., Ltd., Santa Clara, CA, USA) using the standard MIDI (Microbial Identification System) and Sherlock software 6.1 [55].

### 2.7. Symbiotic Characteristics

Cross nodulation tests were carried out for strains SWF67558^T^ and HBU65207^T^ as previously described [16]. The host plants used in the assays were *Medicago sativa* L., *Vigna unguiculata* (Linn.) Walp., *Leucaena leucocephala* (Lam.) de Wit, *Pisum sativum* Linn., *Phaseolus vulgaris* L., *Glycine max* (Linn.) Merr., *Trifolium repens* L., and *Astragalus sinicus* L., and the non-inoculated controls were also included. Eight plant seeds were sterilized in 3.5% (*w*/*v*) sodium hypochlorite and rinsed with sterile water. They were germinated at 28 ℃ until the seedlings grew roots of 0.5–1.5 cm in length; then, the seedlings were inoculated separately via 0.2 mL YM broth cultures of each tested strain (about 10^4^ cells mL^−1^). Two seedlings were transplanted into a sterilized glass tubes (35 × 300 cm) system with sterilized nitrogen-free plant nutrient solution (Vincent, 1970) in 0.8% agar. Five replicates were designed without inoculation and the blank controls were included also with five replicates. The tubes with plants were placed in an artificial climate cabinet (Jiang Nan Instrument, Ningbo, China) under conditions described previously (8 h at 25 °C with light of 12,000 Lx and 16 h at 16 °C in dark) [53].

Moreover, the symbiosis genes responsible for nodulation and nitrogen fixation (*nodA* and *nifH*) were extracted from genome sequences of the four strains and related sequences were searched by blasting in GenBank. All of the *nodA* and *nifH* genes acquired in this study and from the database were used for the reconstruction of the phylogenetic trees as mentioned above for the 16S rRNA genes.

### 2.8. Pigment Identification

Since the four novel *Microvirga* strains presented colonies with pink color, the pigment was extracted and characterized for strain SWF67558^T^. The pure bacterial culture (20 mL) of the tested strains in the TY broth was pelleted by centrifugation. The cell pellet was digested in 10 mL of hydrochloric acid (3M) at room temperature for 1 h, then in boiling water bath for 4 min. The digests were rapidly cooled on ice and centrifuged, and the precipitation was extracted with 20 mL of cold acetone, as previously described [56]. The absorption spectrum of the extract was then recorded between 400 and 600 nm using an Evolution 201 UV-Visible Spectrophotometer (Thermo Fisher Scientific, Inc., Waltham, MA, USA), and the total cellular carotenoid content was determined by absorbance on 503 nm. The acetone phase was reduced by vacuum distillation and then partitioned into an equal volume of hexane. The hexane phase was reduced as previously described for the acetone, and 5 mL of KOH (10%, *w*/*v*) was added and stirred for 20 min in dark; then, gradient-distilled water was added, and the carotenoid phase was applied to the top of a silica gel column; finally, the extract was isolated and purified via Sephadex LH-20 (GE Healthcare, Swiss, Basel, Switzerland). The residue was reconstituted with either an HPLC system solvent (HITACHI DAD HPLC, Hitachi, Ltd., Tokyo, Japan) or hexane for TLC (thin-layer chromatography) separation. The TLC was conducted on high-performance, thin-layer chromatography silica gel plates (from Merck, Rahway, NJ, USA) with hexane-ethyl acetate (4:1, *v*/*v*) as the eluent. The HPLC was used to analyze the extract of strain SWF67558^T^, with column, 5 μm reverse Waters symmetry C18 (150 by 4.6 mm, Alltech, Nicholasville, KY, USA); eluent, acetonitrile (100 *v*/*v*); flow rate, 0.48 mL/min.; detection, 488 nm. For carotenoid isomer identification, a Waters ACQUITY UPLC -MS system (Waters Corp., Milford, MA, USA) was used with 5 μm of reverse Waters symmetry C18 (150 by 4.6 mm, Alltech) and acetonitrile/distill water (100 *v*/*v*) as the eluent, at a flow rate of 0.4 mL/min.

### 2.9. Antioxidant Assays

The radical scavenging activity of the carotenoids extracts was evaluated using DPPH (1,1-diphenyl-2-picrylhydrazyl), H_2_O_2_/Fe^2+^ for the hydroxyl radical system, and superoxide anion radical system, as per the method of Blois [57,58,59]. Accordingly, 200 μL of the carotenoid extract were added to 1 mL of DPPH (0.2 mM in ethanol), and β-carotenoid was used as the standard (60 to 160 μg/mL) for comparing the radical scavenging activity with the extract. The reaction mixture was swirled and placed at 37 °C for 45 min, and its absorbance at 517 nm was checked using the Synergy-HTX (V1.8.8, CioTek). For hydroxyl radicals test, 200 μL of the carotenoid extract were added to 200 μL FeSO_4_ (90 mM), 200 μL salicylates (9 mM distill with ethanol), and 200 μL H_2_O_2_ [60]; the reaction mixture was vortexed and incubated at 37 °C for 30 min, followed by absorbance determination at 510 nm. The superoxide anions were detected by mixing 450 μL of phosphate buffer (0.05 mM, pH 6.6), 100 μL of the carotenoid extract, and 50 μL of Pyrogallol (2.5 mM). After incubation for 5 min, one drop of HCl (8.0 mM) was added, and the absorbance at 320 nm was recorded. Also, β-carotenoid was used as the standard (60 to 160 μg/mL) for comparing the hydroxyl radicals and superoxide anions clearances with the extract.

### 2.10. Reducing Power Assay

The reducing power of the carotenoid extract was determined with the described procedure [61]. Briefly, 200 μL of the extract (60 to 160 μg/mL) was diluted with 200 μL of 0.2 M phosphate buffer (pH 6.6) and 200 μL of potassium ferricyanide (1% *w*/*v*). The mixture was laid at 50 °C for 30 min, then 200 μL of 10% (*w*/*v*) trichloroacetic acid was added and centrifuged at 3000 rpm for 10 min. An aliquot of 500 μL of supernatant was then mixed with 50 μL of ferric chloride (1% *w*/*v*) and 400 μL of distilled water. After 10 min of incubation at 25 ℃, the absorbance was recorded at 700 nm spectrophotometer (Thermo Fisher Scientific, Inc., Waltham, MA, USA). β-carotenoid at the concentration of 60 to 160 μg/mL was used as the standard for comparing the reducing power with the extract.

## 3. Results

### 3.1. Phylogenetic Analyses of 16S rRNA Gene, MLSA with Five Housekeeping Genes and Fingerprints of the Rhizobial Isolates

In this study, the full length 16S rRNA genes of the four tested strains extracted from their genome sequences were 1483 bp for HBU65207^T^ and 1487 bp for HBU67558^T^, HBU67655, and HBU67692. In the phylogenetic trees of 16S rRNA genes (Figure 1 and Appendix A), the four tested strains clustered within the genus *Microvirga*, which presented 99.6–100% of similarities among them and showed the highest similarity to *Microvirga tunisiensis* LmiM8^T^ (99.1–99.3%), followed by type strains for *Microvirga aerilata*, *Microvirga makkahensis*, and *Microvirga vignae* (97.7–98.6%). These results clearly affiliate them as a member of *Microvirga*.

In the phylogenetic tree of the concatenated sequences of *atpD*, *recA*, *dnaK*, *gyrB*, and *glnA* (Figure 2), the four tested strains formed two groups (one with HBU 65207^T^ and HBU 67692; and another with HBU 67,655 and SWF 67558^T^) with sequence similarities lower than the suggested species threshold (97%) with each other and with all the defined *Microvirga* species, demonstrating that these four tested strains represented two novel species within this genus. Furthermore, the BOXAIR profiles of these four strains and *Microvirga tunisiensis* LmiM8^T^ were different from each other (Appendix A), demonstrating that they were not clones of the same strain.

### 3.2. Genome Analyses

The genome sequences acquired from the four tested strains presented sizes between 6.99 and 9.19 Mb with G+C contents of 63.7–64.6%, which were similar with those of the reference strains for the defined *Microvirga* species (Table 1). Among these tested strains, five and six plasmids were detected in HBU67558^T^ and HBU67558^T^, respectively. The gene length/genome and number of protein-coding genes in the genome of the tested strains were also similar with those of the reference strain (Table 1).

The genome properties and distribution of genes into COGs functional categories are shown in Figure 3 (also Appendix A). The distribution of genes was similar in most of the COG categories but not identical in all the five compared genomes (Figure 3). The tested strains SWF67558^T^ and HBU65207^T^ shared very similar numbers of genes distributed in each COG category in the genomes, but strain HBU65207^T^ possessed a higher number of genes than SWF67558^T^ did in most of the COGs categories. There are different significant numbers of clusters of orthologous groups between the tested strains (SWF67558^T^ and HBU65207^T^) and the type strains for *M. tunisensis*, *M. galopogonii*, and *M. lotononidis*, such as transcription, replication, recombination and repair, and energy production and conversion (code of K, L, and C). In the genomes of SWF67558^T^ and 65207^T^, 21 and 26 PKS/NRPS were predicted, respectively (Appendix A). Flagellum synthesis genes *fliP* and *flhAB* were detected in SWF67558^T^, and *flgA*, *fliR*, and *flhAB* were detected in HBU65207^T^, demonstrating that they may move by flagella. All of these data supported the estimation that the tested strains SWF67558^T^ and HBU65207^T^ represent two genomic species different from the defined *Microvirga* species.

In the comparative study of the genomes, the four tested strains were also subdivided into two clusters (Table 2). Strains SWF67558^T^ and HBU67655 were in a group sharing 99.96% ANI and 99.5% relatedness in dDDH; HBU65207^T^ and HBU67692 formed another group, which shared 97.92% ANI and 80.8% dDDH relatedness. The ANI and dDDH values between the two novel groups were 92.12–93.89% and 46.8–53.7%, respectively, while the two values between the two novel group and the reference strains for defined *Microvirga* species were 79.18–86.13% in ANI and 22.9–32.3% in dDDH, respectively (Table 2, Appendix A), which are apparently lower than the species threshold of 94–95% for ANI and 70% for DDH similarity. Hence, it was clear that the four tested strains represented two novel genomic species in the genus *Microvirga*. Moreover, the average amino acid identity (AAI) values were 98.21% between HBU65207^T^ and HBU67692, 99.99% between SWF67558^T^ and HBU67655 (Table 2, Appendix A), and 93.31–93.96% between the two novel groups. While the four strains presented an AAI of 88.13–88.49% for reference strain *M. calopogonii* CCBAU65841^T^, followed by 86.46–87.37% for *M. tunisiensis* LmiM8^T^ (Table 2), and 70.3–88.49% for all of *Microvirga* reference strains with similarity.

The AGIOS (average genomic identity of orthologous gene sequences) values ranged from 84.92 to 90.4 among the compared *Microvirga* species (Table 2). The tested strains SWF67558^T^ and HBU65207^T^ showed AGIOS values ranging from 89.94 to 89.2% for *M. lotononidis* to 90.4–90.1% for *M. tunisiensis*. Meanwhile, the AGIOS value between the tested strains SWF67558^T^ and HBU65207^T^ was 95.4%.

The phylogeny tree of 120 core gene sequences (Figure 4) revealed that the four tested strains were also divided into two groups, with 99.4% similarity between HBU65207^T^ and HBU67692, 99.2% similarity between SWF67558^T^ and HBU67655, and 95.0–95.1% similarity between the two novel groups. Meanwhile, they showed similarities of 90.6–91.5% with the most related reference strain *M. tunisiensis* LmiM8^T^, followed by 83.7–88.8% with *M. calopogonii* CCBAU65841^T^. While similarities between 70.3% and 88.49% were detected among the reference (type) strains of *M. tunisiensis*, *M. calopogonii*, and *M. lotononidis* (Table 2).

### 3.3. Symbiotic Characteristics

Symbiosis genes (*nod* and *nif* clusters) were detected in all four of the tested strains, and they formed effective nodules with *S. cannabina*, but could not nodulate on any of the other legume species involved in the inoculation tests. The phylogenetic trees of symbiosis gene *nodA* (Figure 5) and *nifH* (Appendix A) revealed that the four tested *Microvirga* strains formed a unique cluster with almost identical sequences. In the *nodA* phylogeny (Figure 5), they were most related to the reference strain *M. calopogonii* LMG25488^T^, which was isolated from *Calopogonium mucunoides* grown in Yunnan Province and then clustered to the rhizobia while nodulating with other *Sesbania* species. In the *nifH* tree, these four strains presented relationships closer to the other *Microvirga* species, but distantly related to the *Sesbania*-nodulating strain *Sinorhizobium alkalisoli* YIC 4027^T^.

### 3.4. Phenotypic Characterization

As shown in Table 3, the colonies of the four novel strains on TY plates were light pink, smooth, circular, and opaque, with a diameter of 1–2 mm after 3–5 days of incubation at 28 °C. The cells were Gram-negative and aerobic, non-spore-forming rods with a size of 0.5–0.9 μm × 1.5–2.0 μm. The growth temperature range was 20–45 °C (optimum 28 °C), and the pH range of growth was 6.0–9.0, with an optimum pH of 6.8–7.0. These strains grew weakly in the medium supplied with 1% (*w*/*v*) of NaCl. The cells of SWF67558^T^ and HBU65207^T^ were motile, with a polar flagellum (Figure 6). Strains SWF67558^T^ and HBU65207^T^ were positive for catalase activity, negative for oxidase and urease activities, and negative for denitrification (nitrate to nitrite). Strain SWF67558^T^ was only resistant to chloramphenicol (up to doses of 300 μg mL^−1^), while HBU65207^T^ was only resistant to streptomycin (up to doses of 300 μg mL^−1^). Both strains SWF67558^T^ and HBU65207^T^ utilized L-arabinose, mannitol, D-glucose, D-arabinose, dulcitol, D-galactose, saccharides, glucose, inositol, inulin, rhamnose, sodium acetate, and sodium citrate, but not D-fructose, saccharides, or succinates as their sole carbon source. L-threonine, L-alanine, L-arginine, L-aspartic acid, D-glutamic acid, L-glutamic acid, L-iso-leucine, L-lysine, and L-serine could serve as their sole nitrogen sources. HBU65207^T^ also used L-proline. Details of the differences between the two novel strains and reference strains for the related species are given in Table 3.

### 3.5. Chemotaxonomic Analyses

Here is a disclosure of the analysis results: The polar lipids of strain SWF67558^T^ contained major amounts of phosphatidylethanolamine (PE), phosphatidylcholine (PC), phosphatidylglycerol (PG), and cardiolipin; otherwise, an ornithine-containing lipid was also detected in moderate amounts; while, HBU65207^T^ contained PE, PG, and cardiolipin (Appendix A). The former lacked the ornithine-containing lipid, and the latter lacked cardiolipin and the ornithine-containing lipid.

The cellular fatty acid composition of strain SWF67558^T^ was different from the related strain *M. tunnisiensis* LmiM8^T^ in several cases: C_18:1_ ω7c 11-methyl as major fatty acid (1.97%) and C_18:0_ 3OH, C_20:2_ ω6 and 9c, and C_20:1_ ω7c as minor contents were found in SWF67558^T^, but they were absent in LmiM8^T^. Strain HBU65207^T^ contained C_18:0_ and C_19:0_ cyclo ω8c as the major fatty acids (5.28% and 6.80%). Strain SWF67558^T^ showed the absence of C_17:0_ cyclo and C_20:0_ iso; while, an absence of C_17:1_ ω6c and C_20:0_ iso was observed in strain HBU65207^T^, the reverse of strain *M. tunnisiensis* LmiM8^T^. The major fatty acids (>1% contents) C_16: 0_, C_18: 0_, C_19:0_ cyclo ω8c, summed feature 2, summed feature 3, and summed feature 8 were similar in both SWF67558^T^ and *M. tunnisiensis* LmiM8^T^, although the contents of C_19:0_ cyclo ω8c and summed feature 3 were relatively low in SWF67558^T^ compared to that in *M. tunisiensis* LmiM8^T^ (Appendix A). The major fatty acids (>1% contents) of strain HBU65207^T^ C_16: 0_ and summed feature 2 were similar with that of SWF67558^T^ and LmiM8^T^; the contents of C_12:0.,_ C_17:0_ cyclo, C_18: 0_, C_19:0_ cyclo ω8c, and C_18:0_ 3OH were relatively high in HBU65207^T^ compared to that in *M. tunisiensis* LmiM8^T^ and SWF67558^T^, but summed feature 3 was obviously lower in HBU65207^T^ (Appendix A) than in *M. tunisiensis* LmiM8^T^ and SWF67558^T^.

### 3.6. Analysis of Pigments

The acetone extract of pigment (carotenoids) obtained from SWF67558^T^ in this study showed the highest peak at 503 nm and a broad peak at 525 nm, which were different from the classical three-peaked spectrum (460, 490, 525 nm) for the carotenoid. The TLC analysis showed that strain SWF67558^T^ produced three pink compounds (Rf, 0.85), and the third compound RN3 was dominant (Appendix A). So, further isolation and purification for the extract might be necessary. The result from the HPLC–MS analysis showed an iso peak at 570.3 nm, and the molecular mass showed a peak at 569.3 nm, which is consistent with zeaxanthin of carotenoids, and it even appeared lemon-colored at low concentration and blood-red lipid at high concentration. In the genome, the carotenoids-related genes *crtBI* were found in all the four strains, with 842 bp for *crtB* in all strains, 1481 bp in SWF67558^T^ and HBU67655, and 1550 bp in HBU65207 and HBU67692 for *crtI*.

The result of the antioxidant test showed that the RN3 (β-carotenoids) extract could inhibit free radicals. Quenching of free radicals with 30.74–54.17% contents was observed, less than that of β-carotenoids (equal to 81.2% of β-carotenoids) but was similar to that of H_2_O_2_/Fe^2+^ for hydroxyl radical and superoxide anion radical, scavenging up to 27.03% and 65.79%, equal to more than 85.7% and 95% of β-carotenoids. In addition, the RN3 extract displayed the ability to reduce Fe^3+^ and increase the activity at 160 μg/mL, equal to 90.6% of β-carotenoids standard sampling.

Based on above results obtained in this study, we propose the four tested strains as two novel species in the genus *Microvirga*, named as *Microvirga sesbaniae* and *Microvirga yunnanensis*, respectively. The descriptions of the new species are given in Table 4 and Table 5.

## 4. Discussion

Currently, diverse *Sesbania*-nodulating rhizobia belonging to *Agrobacterium*, *Azorhizobium*, *Bradyrhizobium*, *Mesorhizobium*, *Neorhizobium*, *Rhizobium*, and *Sinorhizobium* have been described [11,12,13,14,16,17,18,19]. In the present study, it is evidenced that bacteria belonging to genera *Microvirga* (Figure 1, Figure 2, Figure 3 and Figure 4) also are the rhizobia associated with *Sesbania*, which further enlarged the spectrum of rhizobia for plants in this genus.

Previously, about 25 bacterial species were described in the genus *Microvirga*, including seven valid and invalid species of nodule-inducing, nitrogen-fixing microsymbionts [23,28,29,30,31] associated with legumes in the four genera. Although the 98.65% similarity of the 16S rRNA gene sequence has been suggested as the threshold to differentiate bacterial species [47], the incapability of the 16S rRNA gene to distinguish the closely related rhizobial species has been well recognized [62]. In the present study, the four novel strains were closely related to the reference strain *Microvirga tunisiensis* LmiM8^T^, with a similarity of 99.1–99.3%, while similarities to greater than 99.66% of 16S rRNA gene were detected among the four novel strains (Figure 1). However, the taxonogenomics or genome sequence comparison among these four strains and reference strains for defined *Microvirga* species clarified that the four tested strains formed two closely related genomic species different from the defined species in this genus (Figure 2, Figure 3 and Figure 4, Table 2).

In the present study, the genomes of these four strains presented sizes (6.99–9.19 Mb) and G+C contents (63.7–64.6%) (Table 1) within the range of *Microvirga* (3.53–9.63 Mb and 61.1–65.1% in Table 1, respectively) [63]. These four *Microvirga* tested strains contained less or equal tRNAs than those in the reference (type) strains, and the distribution of genes in the COG categories in the tested strains (Figure 3) was also similar with that in the reference strains. These data further confirmed the genus affiliation of the four tested strains. Meanwhile, there are significantly different numbers of clusters of orthologous groups between the tested strains and the type strains, such as transcription, replication, recombination and repair, and energy production and conversion (code of K, L, and C) (Figure 3). And the four tested *Microvirga* strains showed PKS/NRPS less than that in the *M. massiliensis* strain [64]. The AGIOS (average genomic identity of orthologous gene sequences) value was more than 99% between the strains in the same species and from 94.3% to 95.4% between the two novel species, which were greater than that of the defined species. Table 2 further supports the classification of these four strains into two novel species. Also, the ANI and dDDH values between the two novel groups and the reference strains for defined *Microvirga* species (79.18–86.13% and 22.9–32.3%, respectively) (Table 2, Appendix A) were apparently lower than the species threshold of 94–95% for ANI and 70% for DDH similarity [65]. Hence, it was clear that the four tested strains represented two novel genomic species in the genus *Microvirga*.

Previously, it has been evidenced that MLSA is a simple and confidential method to define bacterial species. In the present study, the sequence similarities of the concatenated housekeeping genes between the tested strains and the reference strains for defined *Microvirga* species were lower than the suggested species threshold (97%) [66], and the phylogenetic relationships revealed by MLSA are consistent with those of the ANI/dDDH analyses of the genome sequences and the phylogeny of core genes (120 genes), similar to those reported in other studies [18,23,62,67]. So, combining all the results of the 16S rRNA phylogeny, genome comparison, and phylogeny of the MLSA, it could be concluded that the four novel strains investigated in the present study represented two novel genomic species within the genus *Microvirga.*

For the description of the new bacterial species, the phenotypic and chemotaxonomic features are still necessary. In the present study, the characteristics of the polar lipids of the strain SWF 67558^T^ and HBU65207^T^ were different from the closely related strains *Microvirga calopogonii* CCBAU65841^T^ and *M. flocculans* ATCC BAA-817^T^ [68]. And the presence of the flagellum is different from the non-motile cells of the closely related species *Microvirga lupini* and *Microvirga subterranean* [26,28], since *M. lupini* lacked the ornithine-containing lipid, and *M. subterranean* lacked cardiolipin and the ornithine-containing lipid. The cellular fatty acid composition of tested strain SWF67558^T^ was different from the related strain *M. tunnisiensis* LmiM8^T^. In this study, the obtained fatty acid patterns for the reference strains were, in general, consistent with previous descriptions of *Microvirga* strains, and some minor differences may be due to the different cultivation conditions, since the novel strains SWF67558^T^ and HBU65207^T^ could not grow in a TSB agar.

Although *Sesbania* plants could nodulate diverse rhizobial genera, as revealed in previous reports and in the present study, the *nodA* phylogeny (Figure 5) showed that the tested strains formed a unique lineage most similar with the strain *Microvirga calopogonii* LMG25488^T^, a microsymbiont of *Calopogonium mucunoides.* However, it was clear that the *Sesbania* rhizobia within diverse genera and species harbored highly conserved symbiosis genes (Figure 5), as previously reported [10]. So, it could be concluded that *Sesbania* species are promiscuous hosts for their microsymbionts, like common bean and soybean [7,8]. And the *nifH* tree (Appendix A) demonstrated that the *nod* genes and *nif* genes in these four strains may have different origins, just like the case in *Lupinus* bradyrhizobia [69]. These results implied that (1) gene horizontal transfer might have happened among the *Sesbania* symbiotic bacteria, as reported in many other cases [7,20,35], while the symbiosis genes were acquired before the diversification of the two species represented by the four tested strains; and (2) *Sesbania* plants presented strong preference for the symbiosis gene background of their rhizobia, although they are promiscuous hosts associated with diverse species within eight genera: *Agrobacterium*, *Azorhizobium*, *Bradyrhizobium*, *Mesorhizobium*, *Neorhizobium*, *Rhizobium*, and *Sinorhizobium* [11,12,13,14,15,16].

The genus *Microvirga* included several nodule-inducing, nitrogen-fixing species associated with legumes and more species isolated from other environment, such as healthy human skin (*M. mediterraneenis*), human gut (*M. massiliensis*), and sandy arid soil (*M. makkahensis* and *M. arabica*). The characteristic of producing pink pigment is present in some members of this genus, and it has been reported for the symbiotic species *M. lotononidis* and *M. subterranea*, and the non-symbiotic species *M. aerophila* and *M. aerilata*. The pigment identified with carotenoids was only found in the type strain *M. subterranea* Fail4^T^ [26]. In the present study, the four novel strains presented pink colonies, and the pigments from *Microvirga sesbaniae* SWF67558^T^ were identified as carotenoids, mainly zeaxanthin, via HPLC–MS analysis, which coincides with that of *M. subterranea*. Although the other three strains were not analyzed for identification of pigment, the carotenoid biosynthesis genes *crtBI* were detected in the genomes of all the four strains. So, all the four tested strains could be defined as carotenoids-producing bacteria.

With all the phylogenetic and phenotypic divergency, we proposed these pink-pigmented symbiotic bacteria as two novel species, named *Microvirga sesbaniae* sp. nov. and *Microvirga yunnanensis* sp. nov., with SWF67558^T^ (=KCTC82331^T^=GDMCC1.2024^T^) and HBU65207^T^ (=KCTC92125^T^=GDMCC1.2023^T^) as the type strain, respectively. The species description is summarized in Table 4 and Table 5, respectively, and the distinctive features of these two species with the most related species are presented in Table 3.

## Figures and Tables

**Figure 1 microorganisms-12-01558-f001:**
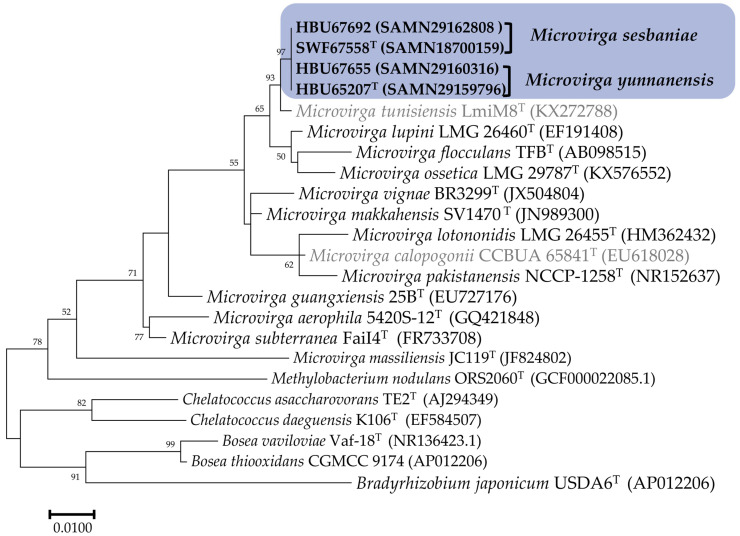
Phylogenetic tree based on 16S rDNA sequences (all the aligned sequences trimmed to 1400 bp) showing the relationships of the rhizobial strains isolated from *Sesbania* and the defined *Microvirga* species. The reference strains marked in gray letter were not validly published. The tree was constructed by using the maximum likelihood method. The bootstrap values > 50% are indicated in the main nodes in a bootstrap analysis of 1000 replicates. The scale bar represents 1% nucleotide substitutions per site.

**Figure 2 microorganisms-12-01558-f002:**
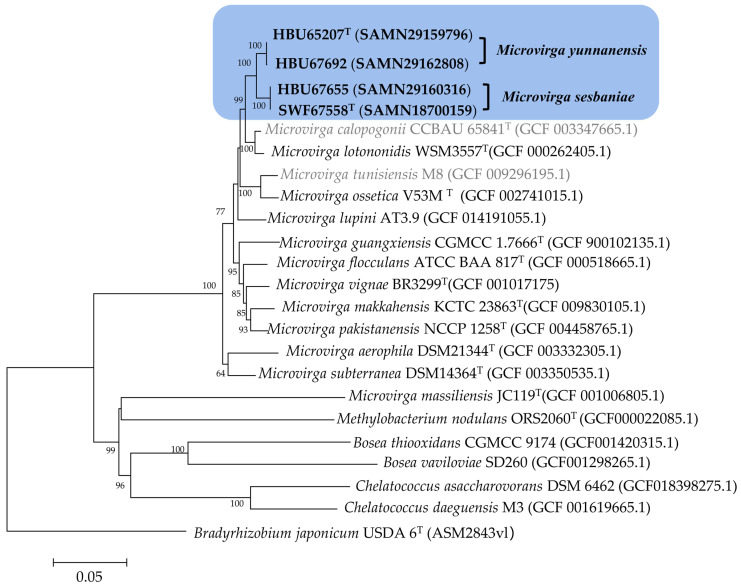
Phylogenetic tree based on the 5 concatenated housekeeping genes (*atpD*, *recA*, *dnaK*, *gyrB*, and *glnA*) sequences (3944 bp) showing the groups of rhizobial strains isolated from *Sesbania* species. The reference strains marked in gray letter were not validly published. The phylogeny was constructed using the maximum likelihood method. The bootstrap values greater than 50% are indicated on the main nodes with a bootstrap analysis of 1000 replicates. The scale bar represents 5% substitutions per site.

**Figure 3 microorganisms-12-01558-f003:**
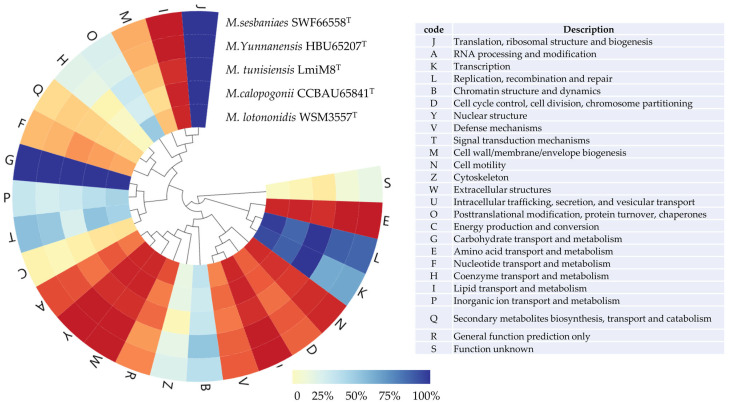
The heatmap of the distribution of the functional classes of the predicted genes in the genomes from *Microvirga sesbaniae* SWF66558^T^, *Microvierga yunnanensis* HBU65207^T^, *Microvirga tunisiensis* LmiM8^T^, *Microvirga calopongonii* CCBAU65841^T^, and *Microvirga lotononidis* WSM3557^T^, according to the COG category.

**Figure 4 microorganisms-12-01558-f004:**
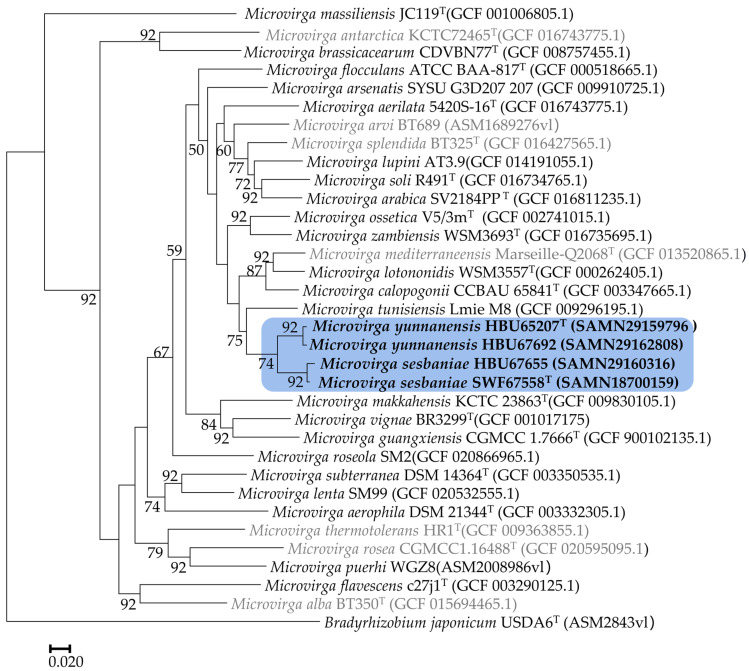
Phylogenetic tree based on 120 core genes extracted from genomes showing the relationships of the rhizobial strains isolated from *Sesbania* with the defined *Microvirga* species. The reference strains marked in gray letters were not validly published. The tree was constructed with the UBCG software. Bootstrap values (>50) are indicated in the main nodes in a bootstrap analysis of 1000 replicates. The scale bar represents 2% substitutions per site.

**Figure 5 microorganisms-12-01558-f005:**
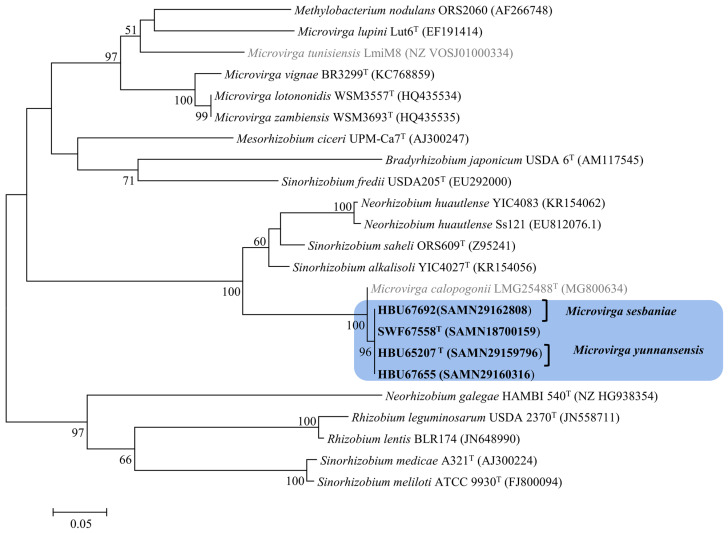
Phylogenetic tree based on *nodA* gene sequences (428 bp) showing the relationships of the Microvirga strains isolated in this study and the other rhizobia. The reference strains marked in gray letter were not validly published. The tree was constructed by using maximum likelihood method. Bootstrap values > 50% are indicated in the main nodes in a bootstrap analysis of 1000 replicates. The scale bar represents 5% nucleotide substitutions per site.

**Figure 6 microorganisms-12-01558-f006:**
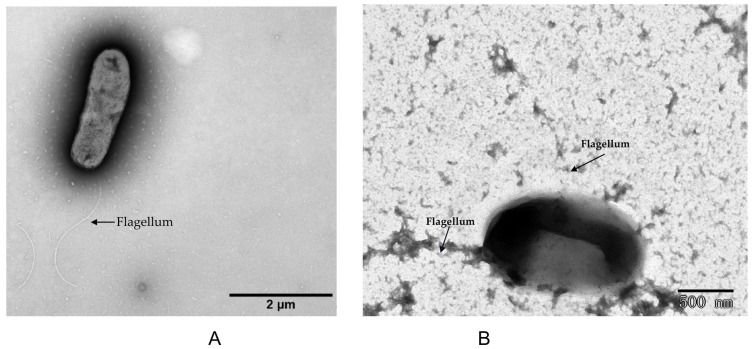
Transmission electron micrograph of the strain *Microvirga sesbaniae* SWF67558^T^ cell with its subpolar flagellum its of strain *Microvirga sesbaniae* SWF67558^T^ (**A**) and *Microvirga yunnanensis* HBU65207^T^ (**B**) grown on TY plate.

**Table 1 microorganisms-12-01558-t001:** General genomic information of the *Microvirga* strains and close related type strains of *Microvirga tunisiensis* LmiM8^T^, *Microvirgacalopogonii* CCBAU65841^T^, and *Microvirga lotononidis* WSM3357^T^.

Strains	Genome Size (Plasmid Numbers)	DNA GC (%)	Gene Length/Genome (%)	The Protein-Coding Genes	RNAs		
					tRNAs	rRNAs	Noncoding RNAs
The tested strains							
***M. sesbaniae* SWF67558^T^**	7.91 Mb (5)	63.69	84.45	7261	67	12	20
*M. sesbaniae* HBU67655	7.22 Mb	63.71	82.42	7146	62	6	7
***M. yunnanensis* HBU65207^T^**	9.19 Mb (6)	63.9	80.92	8005	52	5	9
*M. yunnanensis* HBU67692	6.99 Mb	64.49	82.18	6779	60	3	7
The type strains							
*M. tunisiensis* LmiM8^T^	9.2 Mb	61.5	79.47	8519	67	3	5
*M. calopogonii* CCBAU65841^T^	7.4 Mb	62	83.47	6801	93	15	8
*M. lotononidis* WSM3557^T^	7.1 Mb	63	84.38	6541	68	6	4

**Table 2 microorganisms-12-01558-t002:** Average Nucleotide/Amino Identity (ANI, AAI) (%), the average genomic identity of orthologous gene sequences (AGIOS) (%) among genome sequences, and digital DNA–DNA Hybridization (dDDH) (%) of *Microvirga sesbaniae* SWF66558^T^and *Microvirga yunnanensis* HBU65207^T^ with the related type strains in genus *Microvirga* estimated from the genome sequences.

Type Strains	Tested Strains	%dDDH	%ANIb	%AAI	%AGIOS
*M. tunisiensis* LmiM8^T^	*M. sesbaniae* SWF67558^T^	30.1	84.91	87.36	90.4
*M. calopogonii* CCBAU 65841^T^		32.30	85.83	88.4	90
*M. lotononidis* WSM3557^T^		29.7	84.81	87.0	89.94
*M. tunisiensis* LmiM8^T^	*M. yunnanensis* HBU65207^T^	30.1	84.69	86.46	90.1
*M. calopogonii* CCBAU 65841^T^		32.2	86.04	88.49	89.7
*M. lotononidis* WSM3557^T^		30.9	85.27	86.98	89.2
*M. tunisiensis* LmiM8^T^	*M. sesbaniae* HBU67655	29.9	84.91	87.37	84.87
*M. calopogonii* CCBAU 65841^T^		32.2	85.83	88.13	85.83
*M. lotononidis* WSM3557^T^		29.7	84.81	86.83	84.68
*M. tunisiensis* LmiM8^T^	*M. yunnanensis* HBU67692	30.3	84.99	86.91	84.92
*M. calopogonii* CCBAU 65841^T^		32.3	86.13	88.49	86.06
*M. lotononidis* WSM3557^T^		30.3	85.18	87.05	85.08
*M. yunnanensis* HBU65207^T^		53.7	93.89	93.5	95.4
*M. sesbaniae* HBU67655	*M. sesbaniae* SWF67558^T^	99.5	99.96	99.99	99.99
*M. yunnanensis* HBU67692		46.8	92.12	93.96	94.3
*M. sesbaniae* HBU67655	*M. yunnanensis* HBU65207^T^	53.5	93.8	93.31	95.4
*M. yunnanensis* HBU67692		80.8	97.92	98.21	99.2

Note: The accession numbers of the genomes of the reference strain *Microvirga tunisiensis* LmiM8^T^, *Microvirga lotononidis* WSM3557^T^, and *Microvirga calopogonii* CCBAU 65841^T^ used for this test were GCF 009296195.1, GCF 000262405.1, and GCF 003347665.1, respectively.

**Table 3 microorganisms-12-01558-t003:** Phenotypic differences between *M. sesbaniae* SWF67558^T^ and *M. yunnanensis* HBU65207^T^ strains as compared to closely related type strains of *M. tunisiensis* LmiM8^T^.

Characteristic	*M. sesbaniae*	*M. yunnansensis*	*M. tunisiensisa*	
Isolation source	Root nodule	Root nodule	Root nodule	
Colony colour	Orange	Orange	Pink	
Motilty	Polar flagella	Polar flagella	Polar flagella	
Temperature for growth (°C)				
Optimun	28	28	28	
Range	20–40	20–40	20–37	
pH for growth				
Range	7.0	7.0	6.8–8.0	
	4.0–9.0	4.0–9.0	4–12	
Salt tolerance (*w*/*v*) NaCl	0–1.0%	0–1.0%	0	
Antibiotics sensitivity	Cm^R^	Str^R^	nalidixic acid^R^	
DNA(G+C) content(mol%)	64.59	63.9	61.77	
Symbotic nitrogen fixation	Yes	Yes	Yes	
Arabinose	+	+	+	
Mannitol	+	+	+	
D-Glucose	+	+	+	
D-Fructose	-	-	+	
D-galactose	+	+	+	
Saccharose	-	-	+	
Succinate	-	-	+	
Oxidase	-	-	-	
Urease	-	-	-	
Nitrate reduction	-	-	-	

**Table 4 microorganisms-12-01558-t004:** Description of *Microvirga sesbaniae* sp. nov.

Species Name	*Microvirga sesbaniae*
Genus name	*M* *i* *crovirga*
Specific epithet	*sesbaniae*
Species status	sp. nov.
Species etymology	ses.ba′.ni.ae N.L. gen. fem. n. sesbaniae, of Sesbania, referring to the main host of the species
Designation of the Type Strain	SWF67558^T^
Strain Collection Numbers	KCTC82331^T^=GDMCC1.2024^T^
16S rRNA gene accession nr.	MW828649
Genome accession number	RefSeq = SAMN18700159
Genome status	Complete
Genome size (kbp)	7910
GC mol %	64.59
Country of origin	China
Region of origin	Luxi Country, Yunnan Province
Latitude	24°14′ N
Longitude	98°28′ E
Altitude (meters above sea level)	880 m
Date of isolation	12 December 2007
Source of isolation	Root nodule of *Sesbania cannabina*
Sampling date	10 August 2007
Number of strains in study	2
Source of isolation of non-type strains	Root nodule of *Sesbania cannabina*
Description of the new taxon and diagnostic traits	Cells are Gram-negative, non-spore-forming and aerobic, motile with polar flagellum, rod shaped with the size of 0.5–0.9 μm × 1.5–2.0 μm. Grows on TY medium and nutrient broth (NB). Colonies are light pink, circular, smooth, and opaque. After incubation on TY medium for 2–3 days at 28 °C, the size of the colonies is 1–2 mm in diameter. Cells could grow at 20–40 °C and pH 4.0–9.0. Cells grow optimally in the absence of NaCl, weak could be tolerant to 1% NaCl. Catalase and oxidase test are positive, but nitrate cannot be reduced. Tests for hydrolysis of starch is positive. The cells are sensitive to ampicillin, kanamycin, streptomycin, and tetracycline, but resistant to chloramphenicol. The major polar lipids are phosphatidylcholine (PC), phosphatidylethanolamine (PE), phosphatidylglycerol (PG), cardiolipin and ornithine-containing lipid. The major fatty acids (>3%) are C16: 0, C18: 0, summed feature 2 (16:1 iso I / 14:0 3OH) and summed feature 8 (C18: 1ω7c and/or C18: 1ω6c).

**Table 5 microorganisms-12-01558-t005:** Description of *Microvirga yunnanensis* sp. nov.

Species Name	*Microvirga yunnanensis*
Genus name	*M* *i* *crovirga*
Specific epithet	*yunnanensis*
Species status	sp. nov.
Species etymology	yun.nan.en′sis. N.L. fem. adj. yunnanensis, pertaining to Yunnan, a province of China; where the root nodule strains were isolated
Designation of the Type Strain	HBU65207^T^
Strain Collection Numbers	KCTC82331^T^=GDMCC1.2023^T^
16S rRNA gene accession number	MW830153
Genome accession number	RefSeq = SAMN29159796
Genome status	Complete
Genome size (kbp)	9190
GC mol%	63.9
Country of origin	China
Region of origin	Yuanjiang Country, Yunnan Province
Latitude	23°05′ N
Longitude	102°11′ E
Altitude (meters above sea level)	380 m
Date of isolation	12 October 2016
Source of isolation	Root nodule of *Sesbania cannabina*
Sampling date	18 July 2016
Number of strains in study	2
Source of isolation of non-type strains	Root nodule of *Sesbania cannabina*
Description of the new taxon and diagnostic traits	Cells are Gram-negative, non-spore-forming and aerobic, motile with polar flagellum, rod shaped with the size of 0.4–0.9 μm × 1.5–2.0 μm. Grows on TY medium and nutrient broth (NB). Colonies are light pink, circular, smooth, and opaque. After incubation on TY medium for 2–3 days at 28 °C, the size of the colonies is 1–2 mm in diameter. Cells could grow at 20–40 °C and pH 4.0–9.0. Cells grow optimally in the absence of NaCl, weak could be tolerant to 1% NaCl. Catalase and oxidase test are positive, but nitrate cannot be reduced. Tests for hydrolysis of starch is positive. The cells are sensitive to ampicillin, kanamycin, chloramphenicol, and tetracycline, but resistant to streptomycin. The major polar lipids are phosphatidylethanolamine (PE), phosphatidylglycerol (PG) and cardiolipin. The major fatty acids (>3%) are C16: 0, C18: 0, C19:0 cyclo ω8c, summed feature 2 (16:1 iso I / 14:0 3OH) and summed feature 8 (C18: 1ω7c and/or C18: 1ω6c).

## Data Availability

All data are contained within the article.

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
