# Peer review of "Microvirga sesbaniae sp. nov. and Microvirga yunnanensis sp. nov., Pink-Pigmented Bacteria Isolated from Root Nodules of Sesbania cannabina (Retz.) Poir."

_microorganisms, 2024, doi:10.3390/microorganisms12081558_

Round 1
Reviewer 1 Report
Comments and Suggestions for Authors
In my opinion, the present study of isolates of the genus Microvirga with the characterization of two new species of symbiont bacteria is of interest. I think that the study is well carried out and that conclusive results are obtained. However, the manuscript requires the inclusion of improvements prior to publication. I think that at a background level it can be improved, and a separate discussion of the results is desirable. The results should justify those results that are not shown, improve some legends, and include the statistics in Fig. S6. At the methodology level, it requires the inclusion of some minor details. More detail is indicated in the attached document.

Author Response
Answer : We have deeply checked the manuscript, and improved the background level, added more references and rewrite the context.
Also we have rewritten the discussion, and according to reviewers comments, removed the sentence that belong to the discussion, and placed it on the discussion. In addition, we modified the methodology, added the instrument manufacture, including country, city, and company. And rewrite the phenotypic characteristics, modified the procedures of cross nodulation test. And we agree with the comment that removed the Fig.S6, and thank you for pointing this out.
And other minor details in text modified as follows:
- Line 1, we modified the title and improved by application relevant.
- Line 22, Abstract
We have stated a sentence that include the research and the methods in briefly at beginning. and the total abstract is about 215 words.
- Line 47, we added the references of sentence of Rhizobia are soil bacteria ……..
- Line 107, The sampling site was located in Santaixiang town of Manshi City (N24°19′, E98°24′, 870 m ) and Yuanjiang Town of Yuxi City (N23°35′, E101°53′, 342 m) in Yunnan Province, southwest region of China. And added the species name of sesbania.
- Line 119, we changed the BGI as BGI-Beijing,
- Line 164, we added the references of the Needleman–Wunsch 164 algorithm global alignment technique.
- Line 168, we added the Genbank database as GenBank database (http://www.ncbi.nlm.nih.gov/genbank/).
- Line 185, we added the manufacture, city and country of Mega 7.0 software.
- Line 187, we showed the repetitions.
- Line 192, we added the manufacture, city and country of Biolog system.
- Line 212, we added the manufacture, city and country of GC system.
- Line 217-219, we completed the procedures of cross nodulation test.
- Line 225-227, We removed the sentence of “Since the four novel strains of Microvirga ……”
- Line 233, We added the manufacture, city and country of spectrum.
- Line 246, We added the manufacture, city and country of HPLC-MS system
- Line 277, we made a mistake, there no “data not shown”.
- Line, 291-292, we removed the sentence of “ So, it is clear……..”.
- Line 301-303, we removed the sentence of “Although 98.65% similarity…………..recognized [51]”
- Line 307, we removed the sentence of “both were in…. respectively) [52]”
- Line 312 , we modified the stains information in Table1 legend.
- Line 336, we removed the sentence of “which were …..”.
- Line 364, thank you for point out this, we add the units in Table 2.
- Line 380, we moved the sentence of “Currently, MLSA……MLSA[56–58].”to methods writing.
- Line 430, we corrected the word of “rhizobia” as “rhizobia”.
- Line 438, we changde the word of “are” as “were”.
- Line 456, we added the strain numbers in Table 3.
- Line 466, we moved the sentence of “The characteristic ….” to methods and discussion writing.
- Line 528, we have rewritten a discussion and placed behind the Tables.

Reviewer 2 Report
Comments and Suggestions for Authors
the manuscript entitled<Microvirga sesbaniae sp. nov. and Microvirga yunnanensis sp. 2 nov., isolated from root nodules of Sesbania species> presents interesting information on a new species of the plant Sesbania in China, Authors cultivated and characterized the strain,
Abstract: Four pigment-producing rhizobial strains nodulating with Sesbania cannabina/
delete: with
line204:--were extracted
Could authors explain more Which were the controls?
--the paper is well presented and analyzed, presenting 6 figures and 4 tables.They show Transmission electron micrograph of a rhizobial strain and its subpolar flagellum, It could be better showed the flagellum in \figure 6 a,b.
Figure 6. Transmission electron micrograph of a cell and its subpolar flagellum its of strain/
please check: Transmission electron micrograph of a cell and its subpolar flagellum its of strain... of a cell or of the bacterium? specify
Author Response
Thank you for your comments, and the answer as follows:
1.Answer: we deleted the “with” in abstract and changed “were extract “as were extracted.
2.Answer: we emphasized the controls in 2.2. BOX-PCR for the Microvirga strains, 2.5. Phenotypic characteristics, 2.9. Antioxidant Assays
And the references strain Microvirga tunisiensis LimM8 for Box analysis as the control in BOX-PCR analysis for the Microvirga strains, and all of the tests were designed by YMA medium as the controls in 2.5. Phenotypic characteristics,
And the blank controls without inoculation were included in 2.7. Symbiotic characteristics, and β-carotenoid was used as the control in 2.9 Antioxidant Assays and 2.10 Reducing Power Assay.
3.Answer: we have renewed the resolution ratio of Fig.6, and marked the flagellum in Fig.6-A and B. in which that showed the flagellum grow in the cell of the bacterium, this could be seen by enlarging the figure.

Reviewer 3 Report
Comments and Suggestions for Authors
Final report microorganisms - 3088134
Microvirga sesbaniae sp. nov. and Microvirga yunnanensis sp. nov., isolated from root nodules of Sesbania species
Based on the phylogeny of the 16S rRNA gene in the current study, four pigment-producing rhizobial strains nodulating with Sesbania cannabina (Retz.) Poir. established a distinct group in the genus Microvirga, which is closely linked to Microvirga tunisiensis LmiM8T, with similarity levels of 99.1-39.3%. These four strains were split into two clusters: 99.66% similarity with the 16S rRNA gene, 92.12–94.0% ANI, and 46.7–53.7% dDDH were shared by HBU65207T, HBU67692 and SWF67558T, HBU67655 (100% similarity with the 16S rRNA gene). The results of the genome average nucleotide (ANI) and digital DNA-DNA hybridization (dDDH) analysis showed that the four strains' values were below the species thresholds for related Microvirga species, with ANI≤86.04% and dDDH≠32.3%, respectively.
Five housekeeping genes (gyrB, recA, dnaK, glnA, and atpD) underwent phylogenetic studies, which revealed that these four strains belonged to a highly divergent lineage within the species. The principal polar lipids for strains SWF67558T and HBU65207T were phosphatidylcholine (PC), phosphatidylethanolamine (PE), phosphatidylglycerol (PG), and cardiolipin, with the exception of phosphatidylcholine (PC). Similar major cellular fatty acids, such as C16:0, C18:0, summed features 2 and 8, were present in both strains SWF67558T and HBU65207T, although with differing quantities.
The four strains of HBU67692, HBU65207T, and HBU67558T exhibit different BOX Air characteristics. Zeaxanthin was found to be the pigment extract from strain SWF67558T, and it demonstrated both antioxidant and reduction power. Two novel species, Microvirga sesbaniae sp. nov. and Microvirga yunnanensis sp. nov., were proposed based on phylogenetic and phenotypic dissimilarity. The type strains were SWF67558T (=KCTC82331T=GDMCC1.2024T) and HBU65207T (=KCTC92125T=GDMCC1.2023T).
The material you provided is not well-written and should be read by an English speaker.
Here are some comments you need to make to improve the paper:
Avoid using acronyms in the figure captions or footnotes; instead, ensure that each figure is understandable on its own. Each figure must be self-explanatory.
Please make sure that all tables are self-explanatory and that any abbreviations are explained fully. Despite the fact that all of the references are appropriate, I would include further references from 2023 and 2024.
Looking over the journals, I noticed that some were written in their fullest form and others were written with greater condensation. I have accounted for this in the cited work. When writing in a journal style, be sure to follow the standard structure, which includes either the journal's full name or a shorter version of it. Before you begin, read the guidelines that were provided to the writers.
In the process of supplying references for textbooks, it is recommended that the page numbers of the textbook themselves be noted. Additionally, the city in which the journal is headquartered ought to be given as well.
If the authors were the ones who made all of the comments that were mentioned above, then it is possible that this document might be approved with considerable modifications. In order to ensure that all of my recommendations have been taken into consideration, it will be essential for me to make one more attempt at revising the document.
Comments on the Quality of English Language
Extensive editing of English language required
Author Response
Thank you for your comments, answers as follows:
- Answer: we renewed the material and methods, and read by an English speaker …
- Answer: we modified the figure legends. In figure, we added the strains number and the species name, and because the total name in figure 2 is long, and we inserted a code table in figure2.
- Answer: we modified all of the tables and their legends, added the strains number and units.
- Answer: we modified the references, and include one references in 2023.
- Answer: we have totally checked the references, and recorrected the journal name in standard structure.
- Answer: we make sure the pages inclusion of references, but some are published in web of lacking pages, and the city total name are modified.
Thanks for your comments, and we do our best to modified the manuscript.

Round 2
Reviewer 2 Report
Comments and Suggestions for Authors
The manuscript was improved according to suggestions.